# Photodynamic Therapy in Combination with the Hepatitis B Core Virus-like Particles (HBc VLPs) to Prime Anticancer Immunity for Colorectal Cancer Treatment

**DOI:** 10.3390/cancers14112724

**Published:** 2022-05-31

**Authors:** Yang Hao, Zili Gu, Zhenfeng Yu, Timo Schomann, Sana Sayedipour, Julio C. Aguilar, Peter ten Dijke, Luis J. Cruz

**Affiliations:** 1Translational Nanobiomaterials and Imaging (TNI) Group, Department of Radiology, Leiden University Medical Center, Albinusdreef 2, 2333 ZA Leiden, The Netherlands; y.hao@lumc.nl (Y.H.); z.gu@lumc.nl (Z.G.); z.yu@lumc.nl (Z.Y.); t.schomann@lumc.nl (T.S.); s.s.sayedipour@lumc.nl (S.S.); 2Percuros B.V., Zernikedreef 8, 2333 CL Leiden, The Netherlands; 3Center for Genetic Engineering and Biotechnology, CIGB, Havana 10600, Cuba; julio.aguilar@cigb.edu.cu; 4Department of Cell and Chemical Biology and Oncode Institute, Leiden University Medical Center, Einthovenweg 20, 2300 RC Leiden, The Netherlands

**Keywords:** cancer treatment, photodynamic therapy, TLR-based immunotherapies, viral core particles, vaccination, combined therapy strategy

## Abstract

**Simple Summary:**

Photodynamic therapy (PDT) by means of a photosensitizer is a clinically used therapeutic treatment in a variety of cancers. To further improve the anti-cancer efficiency of PDT, combination therapy with immune agents is a promising option. In this study, we used a viral vaccine as the immune therapeutic partner for PDT. We studied the biological properties of single and combined modalities. Our research suggests that combination therapy enhances innate and humoral immunity, improved survival, and generated a long-term memory capacity in the MC-38 murine colorectal tumor model to prevent a recurrence.

**Abstract:**

Photodynamic therapy (PDT), which combines light and oxygen with a photosensitizer to induce reactive oxygen species (ROS)-mediated killing of primary tumor cells, benefits from non-invasive properties and its negligible toxicity to surrounding healthy tissues. In this study, we have shown that the second-generation photosensitizer FOSCAN can be internalized by tumor cells and effectively induce tumor cell death when exposed to laser irradiation in vitro. In addition, these dying tumor cells can be phagocytosed by dendritic cells and lead to their activation and maturation as assessed by in vitro co-culture models. While PDT induces immunogenic tumor cell apoptosis, its application for the treatment of tumors located in deep tissues and advanced malignancies has been limited. In this study, we demonstrate that hepatitis B core virus-like particles (HBc VLPs) can serve as a vaccine to enhance PDT-induced anti-cancer immunity by priming humoral immune responses and inducing CD8^+^ T cell responses. The combination of PDT and HBc VLPs increased the survival rate of MC-38 tumor-bearing mice to 55%, compared to 33% in PDT alone and no tumor-free mice in vaccine alone. Moreover, the combination effectively prevented tumor recurrence in vivo through enhanced immune memory T cells after therapy. Therefore, as both are clinically approved techniques, this combination provides a promising strategy for cancer therapy.

## 1. Introduction

Colorectal cancer (CRC) is one of the most common tumors with a high incidence and mortality rate [1]. Treatment for colorectal cancer has evolved over the past decades, with a subsequent increase in cure rates, and has become one of the most treatable cancers when detected early [2]. In addition to traditional cancer treatments such as surgery, radiation therapy, targeted therapy, and immunotherapy, photodynamic therapy (PDT) has attracted more attention in colon cancer treatment [3]. Mechanistically, PDT as a cancer therapy approach can exert its cytotoxicity mainly through generated reactive oxygen species (ROS) around the location of the photosensitizer in the tumor area [4]. PDT also impairs vascular structures or elicits immunogenic cell death (ICD) to provide antitumor immunity, and thereby prevents cancer progression [5,6]. Extensive preclinical data suggest that PDT can be used in the treatment of colorectal cancer by deploying it closest to the tumor to directly ablate tumor cells without damaging the connective tissue [7]. However, the application of PDT for clinical translation to colorectal cancer as monotherapy or combination therapy has not been well studied, and there are no clearly approved or recommended photosensitizers for the treatment of colorectal cancer. In this study, we used the clinically approved meta-tetrahydroxy-phenylchlorin (mTHPC, trade name FOSCAN^®^), one of the most potent second-generation photosensitizers for PDT with improved pharmacokinetics and a strong absorption peak at 652 nm [8]. FOSCAN is widely used in the treatment of head and neck cancer and is selectively applied to the medication of breast and pancreatic cancer [9,10,11,12]. Moreover, it shows the promising possibility in application to colon cancer [7].

However, the limited tumor penetration depth of the PDT excitation light source, insufficient PDT-induced immunogenicity, and the complex tumor microenvironment limit the efficiency of non-invasive PDT [13]. Emerging clinical data has shown that PDT in combination with immune adjuvants has the opportunity to increase the therapeutic index window and clinical outcomes for early and advanced-stage solid tumors [14]. Various combinations of immune modulators with PDT were reported to improve the therapeutic response against advanced cancer, including immune checkpoint inhibitors, oncolytic viruses (OVs), and other adjuvants [15].

Here, we study the potential of combining PDT with clinically used hepatitis B core virus-like particles (HBc VLPs), which were produced by and isolated from bacteria [16,17,18]. HBc VLPs containing bacterial RNA has been reported to be released into the cell cytoplasm after being taken up, and elicits strong intrinsic adjuvant properties [19]. Besides being widely used in patients with chronic hepatitis B, studies using HBc VLPs as immune agonists are ongoing in preclinical mice models [20,21,22]. HBc VLPs works as a multi-agonist that can activate the toll-like receptor (TLR)2, TLR3, TLR7, and TLR9 to initiate strong innate and adaptive immune responses [23,24,25]. For instance, Petra et al. showed that a low injection dose of HBc VLPs induced high concentrations of serum IgG and prime Th1 immunity in mice [26]. This HBc VLP formulation specifically stimulates the activation of dendritic cells (DCs), interleukin (IL)-12 release from DCs, and interferon (IFN)-γ release from non-immune spleen cells [26,27,28]. Moreover, when the HBc VLPs are taken up by B cells, subsequently, the activated B cells can present antigen epitopes to naïve T cells, which in turn results in the activation of the T cells [29]. In cancer treatment, these immunologic effects of the HBc VLPs make it an attractive option for combination with PDT to mediate anti-tumor immune responses.

In this study, we investigated the potential of combining FOSCAN-based PDT with therapeutic HBc VLPs that comprise several TLR ligands for the treatment of CRC. We explored the biological function of single therapies in vitro, followed by an investigation of the tumor eradication efficiency of our combined strategy and the overall survival rate of MC-38 tumor-bearing mice. Moreover, the immune responses after combination treatment and long-term immunity were examined. This innovative regimen in which the HBc VLPs vaccine was combined with PDT resulted in improved anti-tumor efficiency and improved anti-tumor responses.

## 2. Materials and Methods

### 2.1. Reagents

Meta-tetrahydroxy-phenylchlorin (mTHPC, trade name FOSCAN^®^, synonyms: KW2345, Temoporfin), used in the present study for PDT, was purchased from MedChemExpress (Princeton, NJ, USA), with CAS No.:122341-38-2. HBc VLPs are defined as the nucleocapsid (core, hepatitis B core antigen) of the hepatitis B virus (HBV). Virus-like particles were produced by means of recombinant DNA technology and obtained from the Center for Genetic Engineering and Biotechnology (CIGB, Havana, Cuba).

### 2.2. Cell Culture

Murine colon cancer cells (MC-38 and CT-26) were cultured in an incubator (Panasonic, Kadoma, Japan) at 37 °C and 5% CO_2_ with IMDM (Lonza, Basel, Switzerland) containing 10% fetal calf serum (FCS; Sigma-Aldrich, St. Louis, MO, USA), 2 mM l-glutamine (Gibco, Paisley, UK), 25 mM β-mercaptoethanol (Sigma-Aldrich, USA), and 100 IU/mL penicillin/streptomycin (Gibco, Paisley, UK), hereafter called full medium. All cells were tested for the absence of mycoplasma and mouse antibody production (MAP) before use.

### 2.3. Animals and Tumor Models

All experimental animals were 6–12-week-old female C57BL/6 J mice purchased from Harlan Laboratories (ENVIGO, Horst, The Netherlands) and housed in pathogen-free animal facilities at Leiden University Medical Center (LUMC). The animal experiments were designed according to the code of practice of the Dutch Animal Ethics Committee, under project license AVD116008045, and approved by the Animal Experimentation Committee of the LUMC.

### 2.4. Uptake and Binding of FOSCAN in MC-38 and CT-26 Cells

MC-38 and CT-26 cells were seeded at a density of 7 to 10 × 10^3^ cells per well in 96-well plates (Greiner, Alphen aan den Rijn, The Netherlands). The next day, the culture medium was refreshed with full medium containing either 0.1 µg/mL or 1 µg/mL FOSCAN and incubated at 37 °C for the uptake assay and 4 °C for the binding assay. For the binding experiments, cells were pre-chilled at 4 °C for 4 h before incubation with FOSCAN. At the indicated time points (0 h, 2 h, 4 h, 8 h, and 24 h), cells were washed, collected, and fixed. Thereafter, the geometric mean fluorescence intensity (gMFI) of cells was measured by an LSRII flow cytometer (BD Biosciences, Franklin Lakes, NJ, USA).

### 2.5. Intracellular Fluorescence Imaging of FOSCAN in MC-38 and CT-26 Cells

MC-38 and CT-26 cells were seeded at a density of 1.5 to 2.5 × 10^4^ cells per well in a 24-well plate (Corning, New York, USA) with micro cover glasses. The next day, the culture medium was refreshed with full medium containing 1 µg/mL FOSCAN at 37 °C. After 24 h, the cells were sequentially washed 3 times with PBS, fixed with 1% paraformaldehyde (Sigma-Aldrich, St. Louis, MO, USA) at RT for 15 min, stained with 50 μg/mL anti-CD44-FITC (Invitrogen, Waltham, MA, USA) at 4 °C for 60 min, washed three times with PBS and stained with 0.2 μM 4′,6-diamidino-2-phenylindole (DAPI; Sigma-Aldrich, USA). After washing three times, glass slides were mounted using Mowiol mounting medium (Sigma-Aldrich, USA) and sealed with nail polish, and thereafter imaged by a Leica DM 5000B fluorescence microscope.

### 2.6. PDT Effects of FOSCAN In Vitro

To detect the FOSCAN toxicity to tumor cells without laser excitation, MC-38 and CT-26 cells were seeded at a density of 7 to 10 × 10^3^ cells per well in 96-well plates (Greiner, Alphen aan den Rijn, The Netherlands). The next day, the culture medium was refreshed with full medium containing a concentration range of FOSCAN at 37 °C for 24 h. Subsequently, the cells were washed and refreshed with FCS-free medium for another 24 h, and thereafter 20 μL MTS reagent (Promega, Madison, WI, USA) was added to each well and incubated for 1–4 h at 37 °C before measuring the optical density (OD) values with a spectrum analyzer (Biolegend, Bio-Rad, iMarkset) at 490 nm. FOSCAN-mediated PDT effects were determined by seeding MC-38 and CT-26 cells at a density of 1.5 to 2.5 × 10^4^ cells per well in a 24-well plate (Corning, New York, USA). The next day, cells were treated for two different experiments: (1) The culture medium was refreshed with full medium containing a concentration range of FOSCAN (0.05 µg/mL, 0.1 µg/mL, 0.25 µg/mL, or 0.5 µg/mL) at 37 °C. After 24 h, the cells were washed before irradiation with a 650 nm laser at a fluence rate of 10 mW/cm^2^ for a total fluence of 2.5 J/cm^2^ to investigate the effects of the FOSCAN dose on PDT-mediated killing effects in vitro; (2) The culture medium was refreshed with full medium containing 0.2 µg/mL FOSCAN at 37 °C. After 24 h, the cells were washed before irradiation with a 650 nm laser at a fluence rate of 10 mW/cm^2^ for a total fluence of 0.1 J/cm^2^, 0.5 J/cm^2^, or 2.5 J/cm^2^ to investigate the effects of fluence on PDT toxicity in vitro. For both experiments, the apoptosis of in vitro PDT was determined 24 h post-illumination by flow cytometer analysis after double staining with Annexin V-FITC (Invitrogen, Waltham, MA, USA) and DAPI.

### 2.7. ATP Release by FOSCAN-PDT-Treated Tumor Cells In Vitro

FOSCAN-mediated PDT was performed as described above. In brief, after 24 h of incubation with 0.2 µg/mL FOSCAN, MC-38 or CT-26 cells were illuminated with a 650 nm laser at a fluence rate of 10 mW/cm^2^ for a total fluence of 2.5 J/cm^2^. The supernatants were collected for ATP release measurement 2 h post-PDT. ATP release was determined by luminescence analysis on a SpectraMax ID3 microplate reader (Molecular Devices, San Jose, CA, USA) using a CellTiter-Glo^®^ Luminescent Cell Viability Assay kit (Promega, Madison, WI, USA).

### 2.8. Flow Cytometry Analysis of FOSCAN-PDT-Treated Tumor Cells In Vitro

A FOSCAN-mediated PDT was performed as described in paragraph 2.6. Next, 2 h after PDT, cells were collected for cell surface CRT and HSP70 expression by staining with recombinant Alexa Fluor^®^ 405 anti-calreticulin antibody (Clone EPR3924, Abcam, Cam- bridge, UK) and Alexa Fluor^®^ 488 anti-HSP70 antibody (Clone W27, Biolegend, San Diego, CA, USA) on an LSR-II flow cytometer. HMGB1 expression was explored by staining with anti-HMGB1/HMG-1 antibody (Novus Biologicals, Littleton, CO, USA) on ice for 30 min, washed three times, and stained with recombinant goat anti-rabbit IgG (H + L) secondary antibody, Alexa 568-PE (Invitrogen, Waltham, MA, USA) before analyzing on an LSR-II flow cytometer.

### 2.9. Phagocytosis of DCs

CellTracker Green CMFDA (5-Chloromethylfluorescein diacetate; Abcam, Cambridge, UK)-labeled tumor cells were seeded into a six-well plate one day before treatment. The next day, 0.2 µg/mL FOSCAN was added to cells for 24 h. After 24 h of incubation, MC-38 or CT-26 cells were illuminated with a 650 nm laser at a fluence rate of 10 mW/cm^2^ for a total fluence of 0.1 J/cm^2^, 0.5 J/cm^2^ or 2.5 J/cm^2^. Immediately after PDT, treated or untreated tumor cells were collected, washed, and added to immature DCs in a ratio of 1:5 for 2 h of co-culture. Thereafter, the collected cells were stained with anti-CD11c-APC-eF780 (clone N418, Thermo Fisher, Waltham, MA, USA) for further analysis with an LSR-II flow cytometer. CMFDA and CD11c double-positive cells were represented as DCs with phagocytosis.

### 2.10. Dendritic Cell Activation and Maturation Induced by FOSCAN-PDT-Treated Tumor Cells In Vitro

In brief, 5 × 10^4^ DCs (immature murine DCs D1) [30] were co-cultured with PDT-treated dying cells in a ratio of 1:5 for 24 h. As a positive control, 1 µg/mL of *E. coli* lipopolysaccharide (LPS) was included in the co-culture. The supernatant of DCs was collected to measure IL-12-p40 levels in the culture medium. The cells were collected and stained with the following antibodies as well as DAPI to determine DC activation: anti-CD11c-APC-eF780, anti-CD40-APC (Clone 3/23, Biolegend, San Diego, CA, USA), anti-CD86-PE-cy7 (clone GL1, BD Biosciences, Franklin Lakes, NJ, USA), anti-I-Ab (MHC class II) (Clone M5/114.15.2, Thermo Fisher, Waltham, MA, USA). Analysis was performed on an LSR-II flow cytometer.

### 2.11. Characteristics of HBc VLPs In Vitro

To analyze the average size and zeta potential, the HBc VLP vaccine solution was measured using a Malvern Zetasizer (Nano ZS, Malvern, UK). The morphology of our HBc VLP vaccine was investigated by transmission electron microscopy (TEM). Images of samples were obtained using a Tecnai 12 Twin TEM (FEI Company, Hillsboro, OR, USA) equipped with a OneView Camera Model 1095 (Gatan, Pleasanton, CA, USA) and a voltage of 120 kV.

### 2.12. DC and B Cell Activation Study by HBc VLPs In Vitro

For DC activation, 5 × 10^4^ DCs were co-cultured with various concentrations of HBc VLPs for 24 h. The supernatant of the DCs was collected for IL-12-p40 measurement. The cells were collected and stained with anti-CD11c-APC-eF780, anti-CD40-APC, anti-CD86-PE-cy7, anti-I-Ab, and DAPI, and then analyzed on an LSR-II flow cytometer. For B cell activation, 6–12-week-old female C57BL/6J mice were inoculated on the right flanks with 4 × 10^5^ MC-38 cells in 100 μL PBS. When mice displayed established tumors, the mice were sacrificed to collect splenocytes (n = 3). Next, 1 × 10^6^ splenocytes were co-cultured with various concentrations of HBc VLPs for 24 h. The supernatant of the splenocytes was collected to measure the IFN-γ levels using an IFN-γ Mouse ELISA Kit (Invitrogen, Waltham, MA, USA). Thereafter, the cells were washed three times with phosphate-buffered saline (PBS) and stained with anti-CD45.2-APC-ef780 (clone 104, Thermo Fisher, Waltham, MA, USA), anti-CD3-BV421 (clone 17A2, Biolegend, San Diego, CA, USA), anti-CD19-BV655 (clone 6D5, Thermo Fisher, USA), anti-CD40-APC (clone 3/23, Biologend, San Diego, CA, USA), anti-CD86-FTIC (clone GL1, Thermo Fisher, Waltham, MA, USA) before analysis by means of flow cytometry.

### 2.13. FOSCAN-Mediated PDT in Combination with Therapeutic HBc VLP Treatments In Vivo

On the right flanks, 6–12-week-old female C57BL/6J mice were inoculated with 4 × 10^5^ MC-38 cells in 100 μL PBS. When mice displayed established tumors at around eight days after inoculation, all mice were randomly divided into four groups: control (PBS), FOSCAN-based PDT, vaccine, and PDT in combination with HBc VLPs (COMB). PDT treatment was performed using a standard protocol: first 0.15 mg/kg FOSCAN was administered to tumor-bearing mice by a slow intravenous (i.v.) injection over four to six min per mouse [31]. Next, 24 h post-administration of FOSCAN, the skin surrounding the tumor area was shaved before mice were irradiated under isoflurane anesthesia at a fluence rate of 23 mW/cm^2^ for a total fluence of 20 J/cm^2^ at 650 nm laser. The next day, mice were injected intratumorally with vaccine every other day for a total of four treatments using 0.5 mg/mL HBc VLP solution in a total volume of 30 µL per treatment. Mice were monitored regularly for tumor growth and weight until the end of the experiment.

### 2.14. Analysis of Spleen-Derived Immune Cell Populations after Treatments

The analysis of immune cell populations in the spleen was performed 12 days after tumor inoculation as described previously [32]. The following staining and antibodies were used: anti-CD45.2-APC eFluor 780, anti-CD3-BV421, anti-CD4-Brilliant Violet 605 (clone RM4-5, Biologend, San Diego, CA, USA), anti-CD8α-APC-R700 (clone 53-6.7, BD Bioscience, Franklin Lakes, NJ, USA), anti-CD19-BV655, anti-CD40-APC, and anti-CD86-FTIC.

### 2.15. Blood Analysis for Immune RESPONSES after treatments

Blood analyses were performed 16 days and 23 days after tumor inoculation. We collected 50 µL caudal vein blood from MC-38 tumor-bearing mice and treated with a lysis buffer to remove erythrocytes. Next, cells were washed three times with PBS and stained with anti-CD45.2-APC-ef780, anti-CD3-BV421, anti-CD4-Brilliant Violet 605, anti-CD8α-APC-R700, anti-CD19-BV655, anti-CD40-APC, and anti-CD86-FTIC. Serum analyses were performed using the IgG (Total) Mouse Uncoated ELISA Kit (Invitrogen, Waltham, MA, USA).

### 2.16. Tumor Re-Challenge and Immune Memory Analysis

The MC-38-challenged, but tumor-free mice (following COMB treatment; n = 6), were re-inoculated with 4 × 10^5^ MC-38 cells in 100 μL PBS on the left flanks. Age-matched naïve female mice (n = 5) were used as control and injected with the same amount of tumor cells. Tumor size was measured as described above. At 21 days after the re-challenge, the blood, spleen, and lymph nodes were harvested to analyze the immune memory. For staining, the following antibodies were used: anti-CD45.2-APC-ef780, anti-CD3-BV421, anti-CD4-Brilliant Violet 605, anti-CD8α-APC-R700, anti-CD44-PE (clone IM7, Thermo Fisher, Waltham, MA, USA), and anti-CD62L-APC (clone MEL-14, Thermo Fisher, USA). The population of CD4^+^ T cells and CD8^+^ T cells were further gated into central memory cells (CD44^+^CD62L^+^), the effector memory cells (CD44^+^CD62L^−^), and naïve immune cells (CD44^−^CD62L^+^).

### 2.17. Statistical Analysis

The data were analyzed using the GraphPad Prism Software version.9.0 (La Jolla, CA, USA). The in vitro data was shown as means ± SEM from three independent experiments. An unpaired two-tailed Student’s *t*-test or one-way analysis of variance (ANOVA) was performed for statistical analyses unless otherwise stated. Statistical differences were considered significant at ns: not significant, * *p* < 0.05, ** *p* < 0.01, *** *p* < 0.001, and **** *p* < 0.0001.

## 3. Results

### 3.1. FOSCAN Uptake, Binding, and Localization In Vitro

We first characterized the binding and uptake kinetics of FOSCAN by cancer cells by detecting gMFI. After incubation with 0.1 µg/mL at 37 °C, the FOSCAN signal of MC-38 tumor cells was detectable, but not at 4 °C. At 1 µg/mL, the fluorescence signal increased with the extension of incubation time at both 37 °C and 4 °C (Figure 1A). Similar results were observed on CT-26 cells (Figure 1B). To investigate the intracellular uptake of FOSCAN, fluorescence images of cells were taken 24 h after incubation with 1 µg/mL of FOSCAN. Strong fluorescence of FOSCAN was observed in the cytoplasm of both MC-38 cells and CT-26 cells (Figure 1C). Taken together, these data demonstrate that FOSCAN is slowly taken up by tumor cells throughout 24 h.

### 3.2. Tumor Cell Responses to PDT In Vitro

Next, we analyzed the effect of FOSCAN on the viability of colorectal cancer cells. When MC-38 (Figure 2A) and CT-26 cells (Figure 2B) were incubated with FOSCAN at a concentration up to 1 µg/mL, no obvious cell damage was observed in both cell lines. However, FOSCAN showed significant toxicity to tumor cells when the concentration was higher than 2 µg/mL after 24 h of incubation. Subsequently, with the introduction of PDT in vitro, approximately 81 ± 3% dead (including early apoptotic, late apoptotic, and necrotic cells) MC-38 cells and 63 ± 7% dead CT-26 cells were observed at 0.1 µg/mL FOSCAN, respectively. The FOSCAN-based PDT induced complete cell death when the concentration of FOSCAN was above 0.25 µg/mL, at 2.5 J/cm^2^ in both MC-38 and CT-26 cells (Figure 2C,D). Additionally, PDT-induced cell death was linked to an increase in fluence (Figure 2E,F). We found that the number of dead MC-38 cells in 0.1 J/cm^2^ fluence PDT-treated tumor cells was around 10% of total cells. When the PDT fluence was increased to 0.5 J/cm^2^, a slight increase of dead cells (approximately 15%) was observed, whereas a significant increase to about 75% was observed at 2.5 J/cm^2^ doses PDT (Figure 2G). The same trend was noticed in CT-26 tumor cells (Figure 2H). Together, the results revealed that FOSCAN has no direct toxicity on tumor cells without light irradiation, but that it can effectively cause cancer cell death after exposure to a 650 nm laser.

### 3.3. Immunological Effects of FOSCAN-PDT-Treated Cancer Cells

To analyze phagocytosis in vitro, DCs were incubated with MC-38 and CT-26 tumor cells killed by FOSCAN-mediated PDT. Live, untreated cancer cells or PDT-treated cancer cells were harvested synchronously after PDT treatment and added to DCs. After 2 h of co-culture, we observed that only debris from dying MC-38 cells (Figure 3A) and CT-26 cells (Figure 3B) were effectively engulfed by DCs when compared to the co-cultures with untreated live tumor cells. The percentage of the debris engulfment proportionally increased with applied PDT fluence doses ranging from 0.1 J/cm^2^ to 2.5 J/cm^2^ (Figure 3A,B). In addition, we explored the immunological effects of FOSCAN-PDT-treated cancer cells by analyzing the expression of the co-stimulatory molecules CD40, CD86, and major histocompatibility complex (MHC)-II on the surface of DCs after 24 h of co-culture. Debris from FOSCAN-PDT-treated MC-38 cells induced a robust upregulation of CD40 expression. This level was comparable to the treatment with the positive control treated with 1 µg/mL TLR4 ligand LPS (Figure 3C). We observed an increase in CD86 expression on DCs exposed to debris from dead MC-38 cells or LPS-treatment (Figure 3D). Furthermore, incubation of DCs with tumor cells, which received different doses of PDT, expressed higher levels of MHC-II on their cell surface without a statistical difference (Appendix A). Moreover, we found significantly increased levels of secreted IL-12-p40 in the supernatant of these co-cultures (Figure 3E). We also found the same trend of increased surface expression of these molecules on DCs challenged with the same amount of debris from PDT-treated CT-26 cells, albeit a weaker response than from exposure to debris from MC-38 cells (Figure 3F–H and Appendix A). Our findings revealed that exposing DCs to debris from PDT-treated cancer cells had immunological effects on DCs maturation in vitro, as evidenced by the elevation of co-stimulatory markers and release of the cytokine IL12-p40.

### 3.4. Physical and Biological Properties of HBc VLPs In Vitro

Next, we characterized the physicochemical and biological properties of the HBc VLPs used in this study. For this purpose, we investigated the morphology of the HBc VLPs by means of TEM, which revealed a uniform size (Figure 4A). In addition, the HBc VLPs exhibited an average size of approximately 30–40 nm and an average zeta-potential (ζ) of −4.29 mV (Figure 4B,C). To investigate the biological effect of the vaccine on immune cells in vitro, we incubated DCs with 1 µg/mL, 2 µg/mL, and 5 µg/mL of HBc VLPs for 24 h. As a positive control, we used 5 µg/mL TLR3 agonist polyinosinic–polycytidylic acid (poly (I:C), a mimic of viral double-stranded RNA) [33]. The expression of CD40 (Figure 4D), CD86 (Figure 4E), and MHC-II (Figure 4F) on the surface of DCs strongly increased in a dose-dependent manner when compared to untreated DCs. Incubation with our HBc VLPs triggered IL-12-p40 secretion (Figure 4G) after 24 h of incubation, indicating activation of these DCs. Additionally, the activation of B cells can also be detected by measuring the expression of CD40 and CD86. For this purpose, we isolated splenocytes from MC-38 tumor-bearing mice and incubated them with different concentrations of HBc VLPs for 24 h. The expression of CD40 on the surface of B cells (CD19^+^CD3^−^) significantly increased at a concentration of 10 µg/mL (Figure 4H), whereas the expression of CD86 was not statistically different at the same concentration (Figure 4I). Moreover, 10 µg/mL HBc VLPs could trigger the IFN-γ secretion of B cells (Figure 4J). Together, these results suggest that the HBc VLPs have a favorable size and charge distribution as well as immunostimulatory activities.

### 3.5. The Combination of FOSCAN-PDT Together with HBc VLPs Vaccine Inhibits Tumor Growth and Increases the Survival Rate of Tumor-Bearing Mice

Based on our encouraging in vitro data, a series of in vivo studies were set up to investigate the efficiency of the combination of PDT and HBc VLPs vaccine on therapeutic effects in vivo. Inoculation with MC-38 cancer cells was selected as a mouse model due to its sensitivity to FOSCAN-PDT and high immunogenicity as compared to the CT-26 model according to our in vitro data (Figure 3). After eight days of tumor cell engraftment, mice from different subgroups received either mock treatment (PBS), standalone treatments (PDT or vaccine), or COMB therapy (Figure 5A). As shown in Figure 5B, mice treated with the HBc VLP vaccine had almost the same tumor size as the rapid-growing tumors in the control group. FOSCAN-PDT treatment and the COMB therapy group exhibited significantly delayed tumor growth for the first three weeks after treatment without significant weight loss (Appendix A). Afterward, the tumor either ablated or resumed (Figure 5B). No significant differences regarding inhibitory effects were observed between PDT and COMB treatment. However, the group which received the COMB treatment showed an increased survival rate (55%) compared to the PDT alone group (33%; Figure 5C and Appendix A). This prolonged survival time was associated with higher levels of leukocytes (CD45.2^+^ T cells) and cytotoxic T lymphocytes (CD3^+^CD8^+^ T cells) in the spleen, although the number of helper T cells (CD3^+^CD4^+^ T cells) was not found to differ between groups (Figure 5D–F). Overall, these data indicate that the HBc VLP vaccine itself had no effects on tumor growth inhibition, but could effectively control tumor growth and significantly improve the survival of treated mice in combination with FOSCAN-PDT.

### 3.6. The Combination of FOSCAN-PDT Together with the HBc VLP Vaccine Induces Antitumor Humoral Immunity

To assess whether PDT in combination with the HBc VLP vaccine can augment levels of circulating lymphocytes, we analyzed the blood of treated mice at two different time points. Our results showed that the population of CD3^+^ CD4^+^ T cells increased in all groups 16 days after treatment, but only in the HBc VLPs and combination groups did the increase persist until 23 days after treatment. There are no statistically significant differences between single and combination treatments (Figure 6A). In contrast to CD3^+^ CD4^+^ T cells, the number of circulating CD3^+^ CD8^+^ T cells in the blood showed no difference between mice treated with PBS and all therapeutic strategies at both time points (Figure 6B). A previous study showed that CD4^+^ helper T cells can interact with B cells, leading to B cell activation through the T cell-dependent pathway for the capability of antibodies [34]. To explore this possibility, we explored B cell activation on day 23. No alterations in the total number of B cells were observed among groups (Figure 6C). However, the expression of both CD40 and CD86 significantly increased on B cells in all treatments, while only CD40 expression showed a significant difference between single PDT and combined treatment. Moreover, we measured the total serum IgG concentration on day 23. As shown in Figure 6D, PDT induced a minimal increase in IgG secretion, while vaccine stimulation induced a moderate increase. Strikingly, the intensity of IgG was significantly increased in the PDT-HBc VLPs-treated group compared to PDT alone group. However, when compared to the HBc VLPs-treated group, the difference is not statistical. These data suggest that B cell activation stimulates antitumor humoral immunity in tumor-bearing mice treated with combined PDT together with HBc VLPs.

### 3.7. The Combination of FOSCAN-PDT Together with HBc VLP Vaccine Modulates the Function of Immune Memory Cells to Protect MICE from Tumor Rechallenge

The same tumor cells were injected on the contralateral side two months after initial treatment, and age-matched naïve female mice were used as control. All mice whose tumors had been eradicated after combination therapy of PDT and HBc VLP vaccination rejected the subsequently injected same tumor cells at the opposite flank two months after primary curative treatment, while control naive mice showed exponential tumor growth (data now shown). This indicates the development of systemic memory immunity after initial tumor cell challenge and subsequent eradication. Memory T cells are classified into central memory T cells and effector memory T cells according to their effector function, proliferative capacity, and migration potential [35]. We evaluated the immune memory phenotype in the blood, tumor-draining lymph nodes (dLNs), and spleen of these mice as illustrated in Figure 7A. We did not observe a significant difference in the level of CD3^+^ CD4^+^ and CD3^+^ CD8^+^ cells between survivor mice after initial COMB treatment and naïve control mice in the blood and dLNs after re-tumor challenge (Figure 7B,E), but it did have an increased number of central memory CD8^+^ T cells and effector memory CD8^+^ T cells in the blood (Figure 7C,D) and dLNs (Figure 7F,G), while naïve immune T cells showed no difference compared to the control group. Moreover, there was a significantly higher level of effector memory CD4^+^ T cells in the lymph node of the surviving mice in the COMB group (Figure 7F). The number of splenic CD3^+^ CD4^+^ T cells in the COMB group drastically increased in comparison with control tumor-bearing mice, whereas no significant changes were observed in CD3^+^ CD8^+^ T cells (Figure 7H). However, we found that the number of effector memory CD4^+^ T cells, naïve immune CD4^+^ T cells, and naïve immune CD8^+^ T cells in the survived mice spleen was increased compared to the control as shown in Figure 7I,J. These findings backed up the notion that FOSCAN-PDT in combination with the HBc VLP vaccination caused a long-term memory response by boosting functional memory T cells.

## 4. Discussion

PDT directly kills tumor cells by generating ROS, following a series of immune responses. Unfortunately, these immune responses are not always strong enough to eliminate tumors. Recently, immune therapies were gradually used to enhance PDT-induced immune responses [14,15]. Therefore, in this study, we evaluated a promising combination strategy for advanced colon cancer, comprising PDT and HBc VLPs. To this end, cytotoxicity of PDT, apoptosis-inducing immunity of PDT-induced tumor cell death, and the physical and biological properties of HBc VLPs were investigated in vitro.

Our results demonstrate that FOSCAN has negligible toxicity in the two isogenic colorectal cancer cell lines (MC-38 and CT-26) at a concentration below 1 µg/mL. However, when the three key components of PDT (i.e., photosensitizer, irradiation, and oxygen) were met, FOSCAN-mediated PDT strongly affected tumor cells by inducing apoptosis. This is consistent with the results of Kiesslich et al. about the dark toxicity of FOSCAN in gall bladder cancer and bile duct cancer cells where FOSCAN concentrations higher than 2 µg/mL could lead to obvious dark cytotoxicity after 20 h of incubation [31]. It is worth noting that under the identical PDT conditions, PDT caused higher cell death in MC-38 cells than in CT-26 cells, which implies that tumor cell lines differ in their sensitivity to FOSCAN-PDT. Furthermore, our findings suggest that both dying MC-38 and CT-26 tumor cells may activate DCs by increasing the release of damage-associated molecular patterns (DAMPs) (Appendix A), which is in accordance with the opinion that photosensitizer-mediated PDT might elicit immunogenicity and activate anti-tumor immune responses in vitro [36,37,38,39].

In this study, we showed that our HBc VLP vaccine has favorable size and charge distributions, and can activate murine DCs and B cells. After HBc VLP stimulation, we observed a significant increase of CD40, CD86, and MHC-II on DCs, as well as secreted IL-12-p40. Next, we examined the in vitro activation of B cells by HBc VLPs. Our data shows that HBc VLPs stimulation can lead to obvious upregulation of the B cell surface markers CD40 and CD86 but the negligible level of CD40, which is due to the high basic expression level of CD40 on the B cell surface and is similar to the report from Lobaina et al. [29]. Furthermore, the cytokine secretion (IL-12 and IFN-γ) of antigen-presenting cells (APCs) following stimulation is consistent with prior findings showing the production of inflammatory cytokines was increased in volunteers’ peripheral blood mononuclear cells and DCs one day after HBc VLPs treatment [25,40].

On the basis of immune activation of both vaccine and FOSCAN, our results indicate that in vivo combination therapy exhibited effective tumor growth suppression, prolonged overall survival, and good prevention of recurrence. In mice with tumors of the more sensitive and responsive MC-38 cancer cells, we were able to show that PDT alone slowed tumor growth, but there was no substantial immunological response of CD4^+^ and CD8^+^ T cells in the splenocytes on day 12. This could be related to the well-known restriction that PDT-induced immune responses are insufficient to eliminate the tumor [41,42,43,44]. In addition, the efficacy of combination therapy was more efficient against tumor growth versus both groups, the control and HBc VLPs alone. The reason that HBc VLP vaccination shows a higher efficacy when combined with PDT might be because (1) it can improve the eradication efficiency of MC-38 tumors; (2) HBc VLPs-PDT treatment increases the number of CD45.2^+^ T cells and CD8^+^ T cells in the spleen; (3) combined treatment induced improved humoral responses in the MC-38 tumor model; or (4) the high production of immune memory T cells that resulted after combined therapeutic modality.

Furthermore, we studied the changes in systemic immunity after combination therapy by examining immune cells in the blood. As reported, CD8^+^ T cells are an important component of the PDT-induced immune response. In addition, viral antigens have been reported to induce strong effector CD4^+^ T cell responses [34,42]. This result corroborates our findings that there is a higher level of CD8^+^ T cells rather than CD4^+^ T cells in the spleen on day 12 when innate immune responses are playing the dominant anti-tumor role. And the percentage of CD4^+^ T cells in the blood was further observed to be significantly increased in all treated groups on day 16, while the increase of CD8^+^ T cells in the blood was inappreciable. This may be because, after this time point (eight days post-PDT), humoral immunity starts to play a major anti-cancer role alternatively. We then explored the hypothesis that CD4^+^ T cells can interact with B cells and lead to B cell activation through the T cell-dependent pathway for the capability of antibodies. We did observe a significant increase in the expression of the co-stimulatory molecule CD40 in B cell populations (CD19^+^ CD3^−^), as well as IgG production in the blood of mice receiving the combination therapy. This suggests that our combination therapy induces CD40-mediated antigen presentation, proliferation, and differentiation of B cells and drives plasma B cells to generate antibody isotypes [45,46,47,48]. We also found that B-cell activation in circulating blood was accompanied by higher levels of IgG in the serum of PDT-treated mice 23 days after PDT treatment. This is in line with previous studies reporting that PDT can activate humoral immunity evidenced by increased serum IgG titers and tumor B-cell infiltration [49]. The propensity for enhanced systematic immunity after combination treatment leads to long-term protection from the same type of cancer as shown in Figure 7. Although the number of CD8^+^ T cells in the spleen did not change following tumor reinfection, we were able to demonstrate that more T cells differentiated into naive memory cells. This differentiation is governed by a linear naïve → effector → central pathway [50]. Our findings point to an increase in effector memory T cells in dLNs and circulating blood, which could provide rapid protection against tumor rechallenge. The combined treatment group had a significantly higher level of central memory cells in dLNs, but not in peripheral blood. This is also consistent with the finding that different memory cell populations occupy different niches [51].

## 5. Conclusions

The impact of combination therapy on the complex tumor microenvironment, cancer metastasis, and other cancer types remains to be explored in preclinical animal models. Even though significant efforts are required to further explore the interaction between the immune system and combination treatment, our results offer the first clues towards the potential of combining PDT with viral antigen-based immunotherapy, and we were able to render new data demonstrating its ability to stimulate anti-cancer immunity.

## Figures and Tables

**Figure 1 cancers-14-02724-f001:**
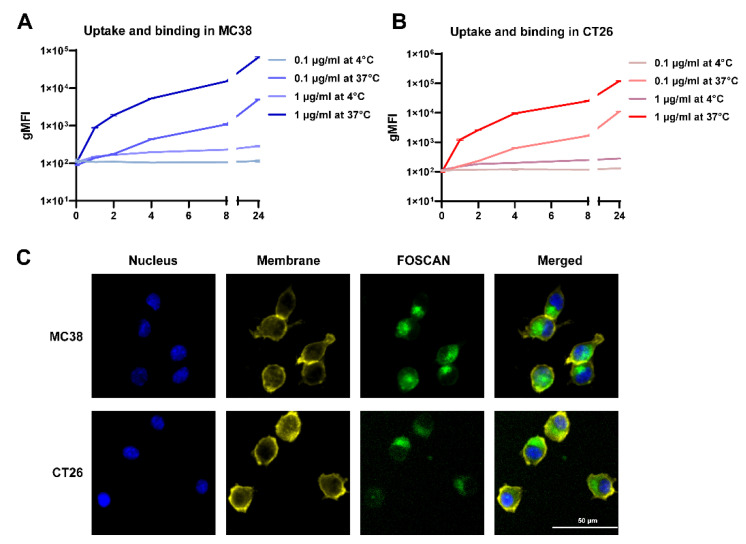
Cellular uptake, binding, and intracellular fluorescence images of FOSCAN in vitro. Uptake (37 °C) and binding (4 °C) assays with either 0.1 µg/mL or 1 µg/mL FOSCAN in (**A**) MC-38 cells and (**B**) CT-26 cells at indicated time points (0 h, 1 h, 2 h, 4 h, 8 h, and 24 h). The graph shows the mean gMFI ± SEM of cells by flow cytometry analysis from three independent experiments. Typical fluorescence microscopy images of (**C**) MC-38 and CT-26 were taken 24 h after incubation with 1 µg/mL FOSCAN. Nuclei (blue), Membrane (yellow), and FOSCAN (green). Scale bar = 50 μm.

**Figure 2 cancers-14-02724-f002:**
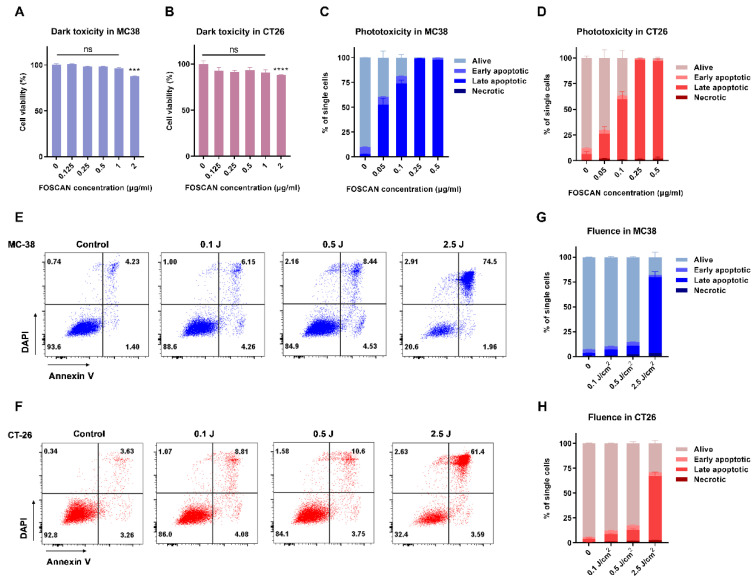
FOSCAN-based photodynamic properties in vitro. (**A**) MC-38 cells (left panel) and (**B**) CT-26 cells (right panel) were incubated with various concentrations of FOSCAN for 24 h, followed by washing and incubation for another 24 h. The next day, MTS assays were performed to determine FOSCAN toxicity without light using a spectrum analyzer. Results were compared by an unpaired Student’s *t*-test and the statistical differences are denoted as ns: not significant, *** *p* < 0.001 and **** *p* < 0.0001. The effect of the FOSCAN concentration in MC-38 cells (**C**), and CT-26 (**D**); the effect of fluence in MC-38 cells (**E**,**G**) and CT-26 cells (**F**,**H**), on the FOSCAN-mediated PDT effects in vitro were investigated by flow cytometry analysis. Cells were incubated with 0.2 µg/mL FOSCAN for 24 h, followed by PDT and incubation for another 24 h. The next day, flow cytometry analyses were performed to determine the apoptosis after double staining with Annexin V-FITC and DAPI. All graphs show the mean values ± SEM from three independent experiments.

**Figure 3 cancers-14-02724-f003:**
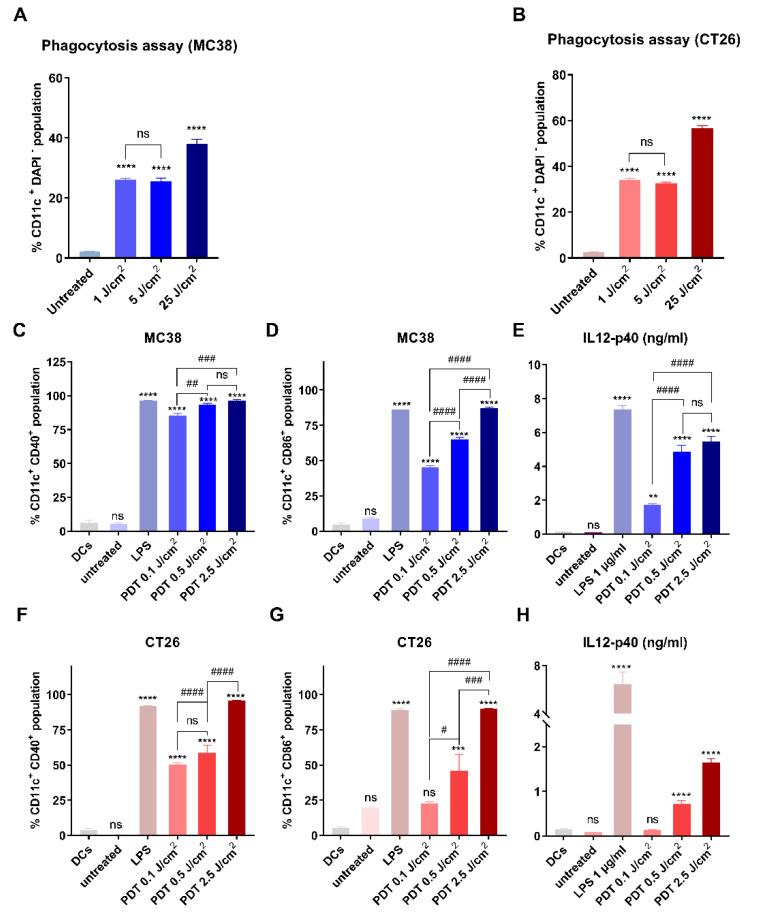
FOSCAN-PDT-treated tumor cells induce phagocytosis and maturation of DCs in vitro. Untreated cancer cells and (**A**) MC-38 or (**B**) CT-26 treated with different doses of PDT were co-cultured with DCs for 2 h immediately post-treatment. Phagocytosis of DCs was measured by determining the percentage of CD11c^+^CMFDA^+^ double-positive DC populations as the mean values ± SEM. Untreated or MC-38 (up), and CT-26 (down) treated with different doses of PDT were co-cultured with DCs for 24 h immediately post-treatment. The percentage of CD40^high^ (**C**,**F**) and CD86^high^ (**D**,**G**) cells in DCs (CD11c^+^DAPI^−^ cells) and IL12-p40 (**E**,**H**) expression in co-culture supernatant were compared to untreated DCs as the control group. All graphs show the mean values ± SEM from three independent experiments. Statistical significance was calculated using the one-way ANOVA and the statistical differences are denoted as ns: not significant, ** *p* < 0.01, *** *p* < 0.001, and **** *p* < 0.0001; # *p* < 0.05, ## *p* < 0.01, ### *p* < 0.001, #### *p* < 0.0001.

**Figure 4 cancers-14-02724-f004:**
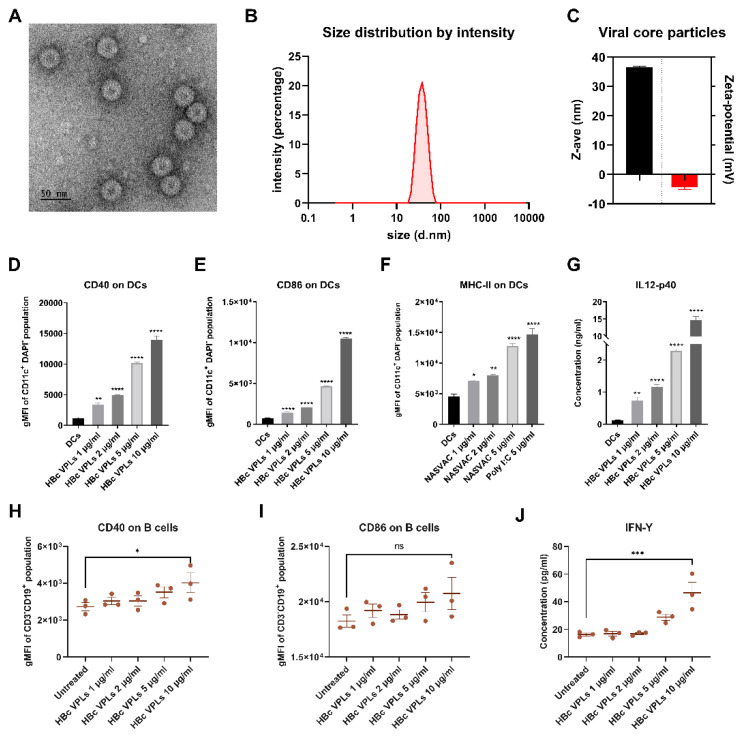
Physicochemical and biological activity of HBc VLPs in vitro. (**A**) Representative TEM image of HBc VLPs (scale bar 50 nm). (**B**) Representative histogram for the size of the viral core particle. (**C**) The average size (z-ave) and zeta potential of the HBc VLPs. (**D**–**G**) In vitro DCs activation assay after 24 h of incubation with viral core particles. The gMFI of CD40 (**D**), CD86 (**E**), and MHC-II (**F**) on DCs (CD11c^+^DAPI^−^ cells) and IL12-p40 secretion (**G**) in the supernatant were compared to the untreated DCs control group. (**H**–**J**) In vitro B cell activation assay using splenocytes from tumor-bearing mice (n = 3) after 24 h of incubation with viral core particles. The gMFI of CD40 (**H**) and CD86 (**I**) on CD45^+^CD19^+^CD3^−^ cells and IFN-γ expression (**J**) the supernatant was compared to untreated control splenocytes. All graphs show the mean values ± SEM from three independent experiments. Statistical analysis was performed using a one-way ANOVA (the statistical differences are denoted as ns: non-significantly, * *p* < 0.05, ** *p* < 0.01, *** *p* < 0.001, and **** *p* < 0.0001).

**Figure 5 cancers-14-02724-f005:**
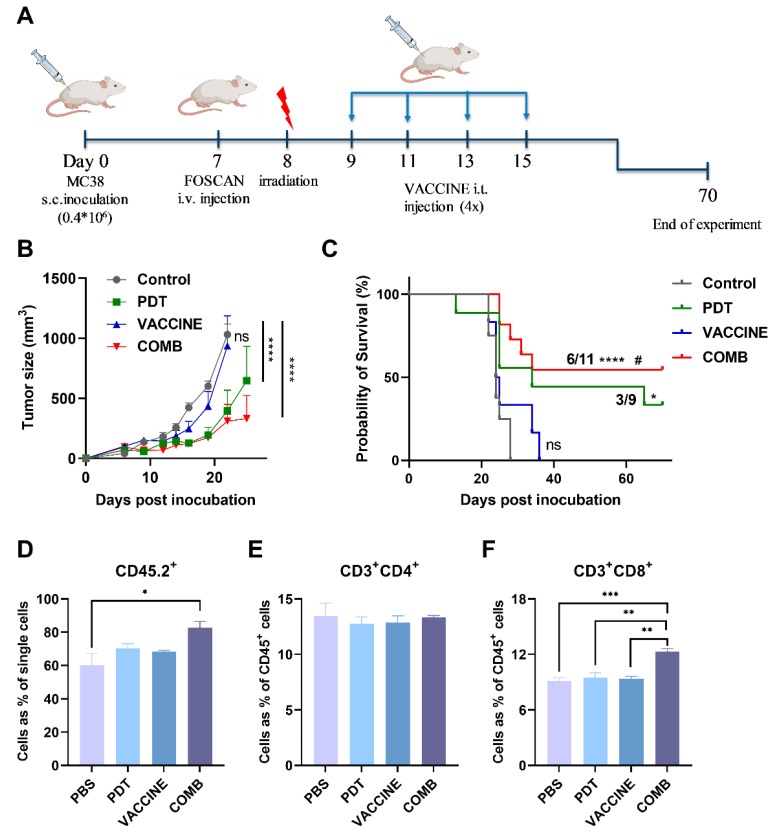
FOSCAN-PDT in combination with HBc VLPs in MC-38 tumor-bearing mice. (**A**) Graphical overview of the experimental design of the MC-38 tumor model, i.t., intratumoral injection. (**B**) The average size of MC-38 tumors of mice after treatment with PBS (Control), FOSCAN-PDT (PDT), HBc VLP vaccine (VACCINE), and PDT combination with the vaccine (COMB). Tumor volume data were presented as means ± SEM and analyzed by a two-way ANOVA. (**C**) The survival times of subgroups by plotting Kaplan–Meier survival curves. Survival analysis was performed with a Log-rank test (* *p* < 0.05, ** *p* < 0.01, *** *p* < 0.001, and **** *p* < 0.0001 by comparing to PBS group; ^#^
*p* < 0.05 by comparing VACCINE to COMB group). One day after the second vaccine injection, mice (n ≥ 3) from different subgroups were sacrificed and spleens were collected, processed, and stained for further analysis of the immune cell population by means of flow cytometry. (**D**) CD45.2^+^ cell population, (**E**) CD4^+^ T cell population, and (**F**) CD8^+^ T cell population. The statistical differences are denoted as ns: not significant, * *p* < 0.05, ** *p* < 0.01, *** *p* < 0.001, and **** *p* < 0.0001.

**Figure 6 cancers-14-02724-f006:**
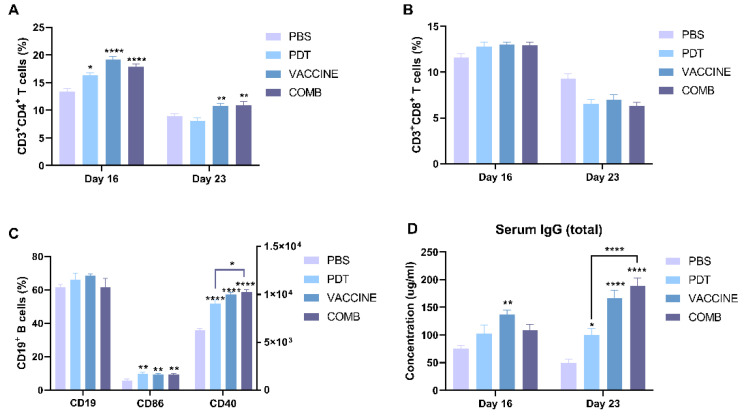
The combination of FOSCAN-PDT and HBc VLPs induces anti-tumor immune responses. (**A**) The percentage of CD4^+^ T lymphocytes and (**B**) CD8^+^ T lymphocytes in the blood of mice injected with different treatments (n ≥ 6). (**C**) The CD3^-^CD19^+^ population and gMFI of CD40 and CD86 on these cells in blood lymphocytes of MC-38 tumor-bearing mice in different subgroups (n ≥ 6). (**D**) Total serum IgG concentrations of MC-38 tumor-bearing mice in different groups (n ≥ 6) were assessed by ELISA. Statistical significance was calculated using a one-way ANOVA. Statistical differences are denoted as * *p* < 0.05, ** *p* < 0.01, and **** *p* < 0.0001.

**Figure 7 cancers-14-02724-f007:**
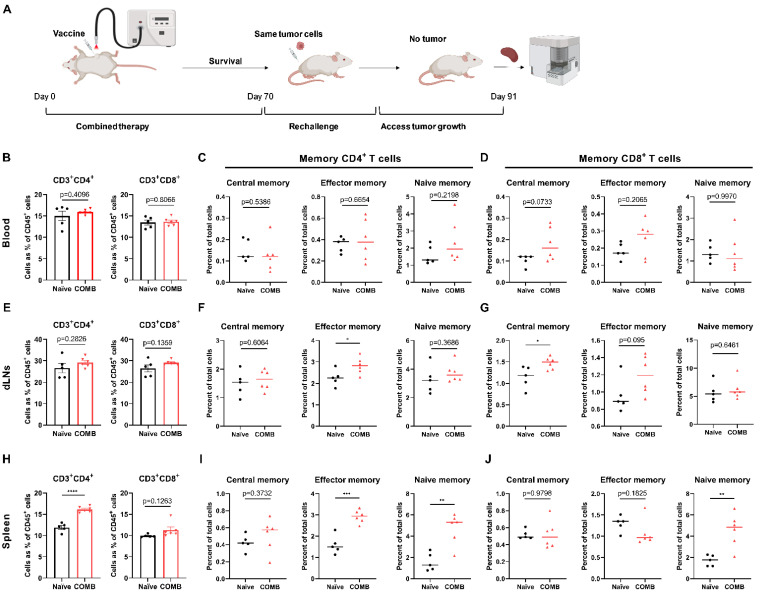
Survival of mice after the treatment of FOSCAN-PDT in combination with HBc VLPs induces immune memory. (**A**) Graphical overview of the experimental design in the MC-38 tumor model. (**B**) The percentage of CD4^+^ T lymphocytes and CD8^+^ T lymphocytes in the blood of mice injected with different treatments at 21 days after rechallenge. (**C**) The percentage of CD4^+^ central memory T cells (CD3^+^CD4^+^CD44^+^CD62L^+^), the CD4^+^ effector memory T cells (CD3^+^ CD4^+^ CD44^+^CD62L^–^), and CD4^+^ naïve memory T cells (CD3^+^ CD4^+^CD44^–^CD62L^+^) in the blood of mice with different treatments at 21 days after rechallenge. (**D**) The percentage of central memory CD8^+^ T cells (CD3^+^CD8^+^CD44^+^CD62L^+^), the effector memory CD8^+^ T cells (CD3^+^CD8^+^CD44^+^CD62L^–^), and naïve memory CD8^+^ T cells (CD3^+^CD8^+^CD44^–^CD62L^+^) in the blood of mice with different treatments at 21 days after rechallenge. (**E**) The percentage of CD4^+^ T lymphocytes and CD8^+^ T lymphocytes in the lymph node of mice injected with different treatments at 21 days after rechallenge. (**F**) The percentage of different memory CD4^+^ T cell populations in the lymph node from subgroups. (**G**) The percentage of different memory CD8^+^ T cell populations in the lymph node from subgroups. (**H**) The percentage of CD4^+^ T lymphocytes and CD8^+^ T lymphocytes in the spleen of mice injected with different treatments at 21 days after rechallenge. (**I**) The percentage of different memory CD4^+^ T cell populations in the spleen from subgroups. (**J**) The percentage of different memory CD8^+^ T cell populations in the spleen from subgroups. Statistical significance was calculated using the Student’s *t*-test, by comparing experimental groups to control (the statistical differences are denoted as ns: not significant, * *p* < 0.05, and ** *p* < 0.01). *** *p* < 0.001, and **** *p* < 0.0001. Black dot represents individual immune cell populations from naïve group and triangle represents individual immune cell populations from COMB group.

## Data Availability

The data presented in this study are available on request from the corresponding author.

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
