# Peer review of "Photodynamic Therapy in Combination with the Hepatitis B Core Virus-like Particles (HBc VLPs) to Prime Anticancer Immunity for Colorectal Cancer Treatment"

_cancers, 2022, doi:10.3390/cancers14112724_

Round 1

Reviewer 1 Report

The work from Cruz and co-workers describes the combined use of foscan-based PDT and hepatitis B core virus-like particles (HBc VLPs), demonstrating that HBc VLPs might serve as a vaccine to enhance PDT-induced anti-cancer immunity by priming humoral immune responses and inducing CD8+ T cell responses.

The study is rigorously designed: two different murine colon cancer cell lines were used for in vitro preliminary studies, highlighting cells' different sensitiveness towards the PDT treatment. Therefore, the authors selected the most sensitive one to perform preliminary in vivo experiments.

Both the experimental plan and the achieved results are convincing and of interest to the readership of the journal, spreading light on the proposed therapeutic combination and deserving additional preclinical studies on more significant in vivo models, e.g., human cancer cell lines in humanised animal models.

Minor comment:

while I understand that the paper is extensive in terms of experiments, data and discussion, the introduction should be slightly enriched. For instance, a mention of why selecting colon cancer for this specific application would be appreciated.  

Author Response

Rebuttal to reviewer comments

We are grateful for the valuable comments by the two reviewers and Editor. By responding to their helpful comments the clarity of our manuscript has greatly improved.

See our point-point response to reviewers’ comments below:

Comments from reviewer #1

Comments: while I understand that the paper is extensive in terms of experiments, data and discussion, the introduction should be slightly enriched. For instance, a mention of why selecting colon cancer for this specific application would be appreciated.

Overall response: In response to the comment by reviewer we have provided motivation for selecting colon cancer for specific application of PDT with HBc VLPs. “Colorectal cancer (CRC) is one of the most common tumors with a high incidence and mortality rate [1]. Treatment for colorectal has evolved over the past decades, with a subsequent increase in cure rates, and has become one of the most treatable cancers when detected early [2]. In addition to traditional cancer treatments such as surgery, radiation therapy, targeted therapies, and immunotherapy, photodynamic therapy (PDT) has attracted more attention in colon cancer treatment [3]. Mechanistically, PDT as a cancer therapy approach can exert its cytotoxicity mainly through generated reactive oxygen species (ROS) around the location of the photosensitizer in the tumor area [4]. PDT also impairs vascular structures or elicits immunogenic cell death (ICD) to provide antitumor immunity, and thereby prevents cancer progression [5,6]. Extensive preclinical data suggest that PDT can be used in the treatment of colorectal cancer by deploying it closest to the tumor to directly ablate tumor cells without damaging the connective tissue [7]. However, the application of PDT for true clinical translation to colorectal cancer as monotherapy or combination therapy has not been well studied; currently, there are no approved PS for the PDT of colorectal cancer. In this study, we used the clinically approved meta-tetrahydroxy-phenylchlorin (mTHPC, trade name FOSCAN®), one of the most potent second-generation photosensitizers for PDT with improved pharmacokinetics and a strong absorption peak at 652 nm [8]. FOSCAN is widely used in the treatment of head and neck cancer and selectively applied to the medication of breast and pancreatic cancer [9,10,11,12]. Moreover, it shows the promising possibility in the application to colon cancer [7].”

Reviewer 2 Report

The authors investigated the potential of FOSCAN-based photo dynamic therapy (PDT) in combination with hepatitis B core virus-like particles (HBc VLPs) for the treatment of colorectal cancer. They explored the biological function of single therapies in vitro, followed by an investigation of the tumor eradication efficiency of their combined strategy and overall survival rate of MC-38 tumor-bearing mice. After combination therapy, they also examined the immune responses and long-term immunity. They concluded that the combined therapy resulted in improved anti-tumor efficiency and improved anti-tumor responses. 

This is a nice study and some data seem clean and are well interpreted. However, better images, clarifications or better explanations are needed in some experiments.

Specific comments are as follows:

  1. Fig. 1: Fig. 1C and D need clarity. CD44-FITC stain in both cells doesn’t seem green. Clear images with FOSCAN (for both cells) are needed to visualize intracellular uptake into the cytoplasm.
  2. Fig. 6: The population of CD3+ CD4+ T cells increased in the treated groups 16 days after treatment (Fig. 6A). In contrast, the number of circulating CD3+ CD8+ T cells in the blood showed no difference between PBS control and the treated groups (Fig. 6B). These results seem contradictory to Fig. 5E and 5F, where cytotoxic T lymphocytes (CD3+CD8+ T cells) in the spleen of the combination drug treated animals are increased compared to the CD3+ CD4+ T cells. This must be explained. Also, PDT (alone)-induced increase of CD4+ T cells (6A), B-cell activation (6B), and serum IgG should be discussed.
  3. Fig. 7: It is not clear from what is represented by the “PBS” (7B-H). From the authors’ description in Section 3.7 “The FOSCAN-PDT combination and the HBc VLP vaccine alone did not affect the CD3+ CD4+ and CD3+ CD8+ cell populations (Fig. 7B, C)”, it seems “PBS” group represents animals that previously received the HBc VLP vaccine. But, in the method section 2.16, authors described as follows: “The MC-38-challenged, but tumor-free mice (following COMB treatment; n=6), were re-inoculated with 4 × 105 MC-38 cells in 100 μl PBS on the left flanks. Age-matched naïve female mice (n=5) were used as control and injected with the same amount of tumor cells.” If the PBS group represents naïve mice, then the results in Fig. 7 should be discussed in this context.
  4. Colorectal cancer affects both males and females, yet only female mice were studied here. Is there any justification for this preference?

Author Response

Comments from reviewer #2

  1. 1C and D need clarity. CD44-FITC stain in both cells doesn’t seem green. Clear images with FOSCAN (for both cells) are needed to visualize intracellular uptake into the cytoplasm.

We are grateful for your suggestions. In the revised manuscript, we have replaced the Fig 1 with new images for better visualization. CD44-FITC staining was used for visualization.

  1. The population of CD3+ CD4+ T cells increased in the treated groups 16 days after treatment (Fig. 6A). In contrast, the number of circulating CD3+ CD8+ T cells in the blood showed no difference between PBS control and the treated groups (Fig. 6B). These results seem contradictory to Fig. 5E and 5F, where cytotoxic T lymphocytes (CD3+CD8+ T cells) in the spleen of the combination drug treated animals are increased compared to the CD3+ CD4+ T cells. This must be explained. Also, PDT (alone)-induced increase of CD4+ T cells (6A), B-cell activation (6B), and serum IgG should be discussed.

We are appreciative of your comments. Different time points must be the reason behind this difference. One day after the second vaccine injection (day 12), we analyzed the immune cells in spleen in order to study the activation of the adaptive immune response (lymphocytes). We found that all treatment induced CD45+ leukocytes in the spleen on day 12. This change may be directly attributable to the acute inflammation and activation of adaptive immunity resulting from PDT treatment, as well as the immune effect triggered by HBc VLP through activation of DCs. CD8+ T cells have been described as key players in PDT-mediated long-term tumor control while CD4+ T cells plays a supporting role [1,2], yet neither PDT nor HBc VLP treatment alone was sufficient to cause cytotoxic T lymphocytes, and only COMB promoted significant recruitment or production of CD8+ T cells in the spleen (Fig. 5E/F). Then we studied circulating immunity on day 16, we found the changes on CD4+ T cells rather than CD8+ cells in the circulating blood. This may be due to at this time point (8 day post-PDT), the PDT-dominated T-cell response is almost exhausted and humoral immunity (CD4 T cells, B cells) plays a major anti-cancer role in the following time (Fig. 6) [3,4,5]. We have include a discussion in the revised text to explain the reason, please see line 531 to 539 (revised manuscript).

In response to reviewers’ comments, we have included a discussion in the manuscript to explain PDT induced increase of CD4+ T cells (6A), B-cell activation (6B), and serum IgG,, please see line 446 to 449 (revised manuscript).

  1. 7: It is not clear from what is represented by the “PBS” (7B-H). From the authors’ description in Section 3.7 “The FOSCAN-PDT combination and the HBc VLP vaccine alone did not affect the CD3+ CD4+ and CD3+ CD8+ cell populations (Fig. 7B, C)”, it seems “PBS” group represents animals that previously received the HBc VLP vaccine. But, in the method section 2.16, authors described as follows: “The MC-38-challenged, but tumor-free mice (following COMB treatment; n=6), were re-inoculated with 4 × 105 MC-38 cells in 100 μl PBS on the left flanks. Age-matched naïve female mice (n=5) were used as control and injected with the same amount of tumor cells.” If the PBS group represents naïve mice, then the results in Fig. 7 should be discussed in this context.

We apologize for the unclear statements. We have revised Fig 7 by changing the group name and modified the discussion in Section 3.7, please see line 451 to 465 (revised manuscript).

  1. Colorectal cancer affects both males and females, yet only female mice were studied here. Is there any justification for this preference?

Many thanks for this valuable comment. Female mice were studied in the present study for a practical reason; male mice tend to fight more in cages than female mice [6]. Fight wounds and additional stress may affect the immune system and therefore the responses to therapy. The age of the mice, at the start of the experiments, was 6 to 8 weeks. At this  age the mice have a fully functional immune system that is representative to the adult human immune systems [7].

References

  • Agostinis P, Berg K, Cengel KA, et al. Photodynamic therapy of cancer: an update. CA Cancer J Clin. 2011;61(4):250-281. doi:10.3322/caac.20114
  • Mroz P, Hashmi JT, Huang YY, Lange N, Hamblin MR. Stimulation of anti-tumor immunity by photodynamic therapy. Expert Rev Clin Immunol. 2011;7(1):75-91. doi:10.1586/eci.10.81
  • Wachowska M, Gabrysiak M, Muchowicz A, et al. 5-Aza-2'-deoxycytidine potentiates antitumour immune response induced by photodynamic therapy [published correction appears in Eur J Cancer. 2021 Jan;142:150-151]. Eur J Cancer. 2014;50(7):1370-1381. doi:10.1016/j.ejca.2014.01.017
  • Beltrán Hernández I, Yu Y, Ossendorp F, Korbelik M, Oliveira S. Preclinical and Clinical Evidence of Immune Responses Triggered in Oncologic Photodynamic Therapy: Clinical Recommendations. J Clin Med. 2020;9(2):333. Published 2020 Jan 24. doi:10.3390/jcm9020333
  • Lobaina Y, Hardtke S, Wedemeyer H, Aguilar JC, Schlaphoff V. In vitro stimulation with HBV therapeutic vaccine candidate Nasvac activates B and T cells from chronic hepatitis B patients and healthy donors. Mol Immunol. 2015;63(2):320-327. doi:10.1016/j.molimm.2014.08.003
  • To Group or Not to Group? Good Practice for Housing Male Laboratory Mice Kappel S et al. Animals (Basel). 2017 Nov 24;7(12). pii: E88. doi: 10.3390/ani7120088.
  • The mouse in biomedical research, James G. Fox (ed.), American College of Laboratory Animal Medicine series (Elsevier, AP: Amsterdam; Boston).

Round 2

Reviewer 2 Report

The authors adequately addressed this reviewer's all concerns.